# A Comparative Study of Serum Pharmacochemistry of Kai-Xin-San in Normal and AD Rats Using UPLC-LTQ-Orbitrap-MS

**DOI:** 10.3390/ph16010030

**Published:** 2022-12-26

**Authors:** Lin Yang, Jian Liang, Qin Zheng, Lifen Zhou, Yongchang Xiong, Huijuan Wang, Jinbin Yuan

**Affiliations:** 1Key Lab of Modern Preparations of Traditional Chinese Medicine, Jiangxi University of Chinese Medicine, Nanchang 330004, China; 2Research Center for Traditional Chinese Medicine Resources and Ethnic Minority Medicine, Jiangxi University of Chinese Medicine, Nanchang 330004, China

**Keywords:** Kai-Xin-San, serum pharmacochemistry, AD-model rat, ultra-high performance liquid chromatography-linear ion trap-Orbitrap mass spectrometry (UPLC-LTQ-Orbitrap MS), multivariate statistical analysis, prototype component, metabolites

## Abstract

Kai-Xin-San (KXS) is a classic formula for the treatment of Alzheimer’s disease (AD). KXS has been widely used to treat emotional diseases; however, its active components remain unknown. There have been some reports about the efficacy and metabolic analysis of KXS, which are mainly based on studying normal animals. The current work first established an AD rat model by injecting D-galactose into the abdominal cavity and injecting Aβ_25–35_ into the hippocampus on both sides, followed by intragastric administration of KXS for a consecutive week; then, the analytical method for ethanol extraction from the serum of normal and model rats was developed using UPLC-LTQ-Orbitrap-MS; finally, the transitional components in the blood were systematically compared and analyzed by multivariate statistical analysis. A total of 36 components of KXS were identified in the rat serum of the normal group, including 24 prototype components (including ginsenosides, triterpenoid acids of *Poria cocos*, polygala saponins, polygala xanthones and polygala ester) and 13 metabolites (including desugar, hydration and oxidation products of ginsenosides, triterpenoid acid hydroxylation, deoxygenation, demethylation, desaturation, and glycine-conjugated products of *Poria cocos*). Twenty KXS-relevant components were detected in the rat serum of the model group, including 11 prototypes and 9 metabolites. The normal group and the model group shared 12 common components, including 9 prototypes and 3 metabolites. The intestinal microecological balance of the model rats probably was destroyed, affecting the absorption/metabolism of saponins by the body, which resulted in fewer transitional components in the model group. This study reflected the drug-body interaction from an objective and accurate perspective, offering references and insights for elucidating the basis of active components and mechanism of action of KXS for treating AD.

## 1. Introduction

Kai-Xin-San was first recorded in “Volume 14 Small intestine of *Essential Recipes for Emergent Use Worth A Thousand Gold*”, which consists of the following 4 drugs: ginseng radix et rhizoma, polygalae radix, poria and acori tatarinowii rhizoma. The sweet and warm ginseng radix et rhizoma is the monarch, nourishing Chi, inducing resuscitation and benefiting the intellect; it is assisted by poria that is mildly sweet, with calming and relaxing properties. The addition of mildly bitter Polygalae radix can relieve fright, benefit the intellect and dispel depression. The mildly spicy acori tatarinowii rhizome can unclog apertures and guide other components to ascend. Based on the above-mentioned characteristics, physicians invented many similar formulas by modulating the ratio or adding more drugs; such examples are Ding-zhi bolus, Bu-xin decoction, and Sang-piao-shao powder [1]. With the variation of the ratios of 4 component drugs, their indications focus on different aspects, which mostly are emotion-related diseases. Recent studies have showed that KXS is efficacious for the treatment and prevention of psychotic disorders such as senile dementia and depression [2,3].

At present, most KXS studies focus on its pharmacodynamics, and there have been in vivo metabolic studies on it and its component drugs. Using UPLC-Q-TOF/MS, Liu identified 6 prototype components and 7 relevant metabolites in rats after the oral administration of polygalae radix [4]. Using ESI-MS, Ling identified 1 prototype component in rat serum after the oral administration of poria [5]. Similarly, Zhang identified 26 prototype components using UPLC-Q-TOF/MS in the biological sample of rats after the oral administration of the aqueous extract of KXS [6]. In the follow-up study, they identified 69 KXS-relevant components in rat serum among the in vivo transitional components [7]. The problem is that these studies are all based on normal animals; as there have been reports that the absorption and metabolism of the same drug in normal animals are significantly different from those in animals in a pathological state, studying the animals in a pathological state is more accurate considering the real drug-body mechanism of action [8,9]. In this sense, it is necessary to discuss the characteristics of absorption and metabolism of KXS under the AD condition, if the martial basis of the active components is to be evaluated accurately.

Combining the classic research approaches of pharmacochemistry and the modern means of chromatography and mass spectrometry, serum pharmacochemistry of traditional Chinese medicine (TCM) explores the active components that really work in vivo, which offers support for the quality-control study of TCM. Given the complex components in TCM, manually analyzing the peaks on a spectrogram will inevitably lead to neglect. Taking multivariate statistics in metabonomics for reference, the transitional components in blood could be characterized rapidly and comprehensively by extracting the ions in pre-treated spectrograms and setting up parameters.

UPLC-LTQ-Orbitrap-MS (ultra-high performance liquid chromatography-linear ion trap-Orbitrap mass spectrometry) is a technique that has many advantages such as high resolution, high mass-accuracy, and wide dynamic range, allowing it to be a powerful research tool for the study of complex TCM components. With the development of modern LC-MS techniques and high efficiency of data-processing platforms, the idea of combining serum pharmacochemistry of TCM can achieve better recognition and allow tracking of the converting process of active TCM components in vivo [10,11]. Following this idea, the current study initially replicated the AD rat model, and verified whether the model was a success via behavioristics and histomorphology; then, using the combination of UPLC-LTQ-Orbitrap-MS, serum pharmacochemistry and multivariate statistical analysis, the transitional components in normal rats and AD-model rats with the oral administration of KXS were systematically compared. The findings in this work could be of help for the study of the material basis of KXS for treating AD. The whole study process is summarized in Figure 1.

## 2. Results

### 2.1. Assessment of the AD Rat Model

#### 2.1.1. General Behavioral Observations

During the modeling, the hair of normal rats had a uniform color and was glossy. The model rats consumed less food and water. The normal rats put on weight at a natural rate, whereas the model rats hardly showed any change in their weight. The normal rats were relatively active, whereas the model rats were inert, inactive and indifferent.

#### 2.1.2. Morris Water Maze Test

As Figure 2 demonstrates, in the orienting-navigation test, AD rats had evident prolonged escape latency (*p* < 0.05) on the 3rd and 4th days and significant prolonged escape latency (*p* < 0.01) on the 5th day. In the space-exploring test, the swimming speed of AD rats and the platform crossover number were significantly lower than those of the normal rats (*p* < 0.05), while the target dwell time was significantly shorter than that of the normal rats (*p* < 0.01). The tracks of model rats were evidently prolonged in the space-exploring test (showed in Figure 2F); the tracks of the normal rats were closer to the platform. The above results of behavioral experiments indicated that the learning ability of rats decreased after modeling, suggesting the AD modeling was a success.

#### 2.1.3. Histomorphology of the Hippocampus

The pathological section of the rat hippocampus is shown in Figure 3. The hippocampal neurons of normal rats were densely distributed in an orderly fashion; the morphology of cells was normal, with nuclei in the center and clear nuclear membranes. In contrast, the hippocampal neurons of model rats were relatively sparse; some neurons were ruptured, Nissl bodies decreased in number, nuclei shrunk and were concentrated, and the boundaries of cells were vague.

### 2.2. Acquisition of Chromatograms of Biological Samples

The ethanol extract of KXS and serum samples of the blank normal group (CK), KXS-treated normal group (CG), blank model group (MK), and KXS-treated model group (MG) were quantitatively analyzed under the conditions listed in Section 4.2.7 and Section 4.2.8. The total ion chromatograms (TIC) in positive and negative modes are shown in Figure 4. As it can be observed in Figure 4(A1,B1), KXS-relevant components achieved good separation within 80 min, and 160 principal components were identified based on our previous work [12,13]; the relevant MS information has been summarized in Appendix A, which could be helpful for the identification of the components in serum. Comparing the TICs in both positive and negative modes (Figure 4A vs. Figure 4B), it could conclude that TIC in negative mode had a better response, which was then selected for the further MS analysis. Significant differences were observed in the intensity and number of peaks on the serum chromatograms of the normal group vs the model group and those before vs after dosing; however, the components with low levels or response would readily be missed if their identification was based on the MS information of these differential peaks. In addition, the response of endogenous components was stronger than that of KXS-relevant components, interfering with the identification of transitional components; hence, the in vivo prototype components and metabolites of KXS were recognized and identified with the assistance of multivariate statistical analysis (PCA and OPLS-DA).

### 2.3. Multivariate Statistical Analysis

Peak-identification, peak-matching and normalization of the original data were processed using the software SIEVE (edition 2.1); then, these data were imported into software SIMCA (edition 14.1). PCA is capable of reducing the dimensions of data, checking up whether there are differences among different groups on a scatter plot; then, the fragment ions with VIP > 1.5, *p* < 0.05 and FC > 2 were screened out using OPLS-DA to explore the differential components. The PCA results of serum of the four groups are presented in Figure 5A: first, the normal group (CK: blank normal group; CG: KXS-treated normal group) and the model group (MK: blank model group; MG: KXS-treated model group) were gathering on both sides of the Y-axis; second, the differences before and after administration were evident, which can be seen that they were gathering on both sides of the X-axis. This phenomenon showed that the four groups could be well distinguished in the current model. In Figure 5B,C of the loading plot of OPLS-DA, the samples before and after administration were divided into two groups, suggesting that there were differential components in the serum before and after administration. Similar results can be seen in the Appendix A. With the above parameters, the in vivo differential components were screened out and 2796 fragment ions were acquired for the normal groups and 1577 fragment ions for the model groups. Appendix A exemplifies the filtered ions and their identified parameters. 

### 2.4. Analysis and Identification of KXS Components in the Blood

The above differential ions described in Section 2.3 need further combing, confirmation and verification. The in vivo prototype components and metabolites were identified by using Compound Discoverer 3.2 based on accurate mass, retention time, fragment ion, literature, as well as reference standards and an in-house database (Appendix A). A total of 24 prototype components were identified in the rat serum of the KXS-treated normal group, 11 in the KXS-treated model group, and 9 were common components (summarized in Table 1). Thirteen metabolites were identified in the rat serum of the KXS-treated normal group, 9 in the KXS-treated model group, and 3 were common components (summarized in Table 2).

### 2.5. Identification of the Prototype Components

A total of 24 prototype components were identified in the serum of the normal group after the administration of KXS, including eight ginsenosides, four polygala saponins, two polygala xanthones, one polygala ester and nine poria triterpenes. Eleven prototype components were identified in the serum of the model group, including five ginsenosides, one polygala saponin and five poria components.

Component P2: a quasi-molecular ion [M−H]^−^ peak was produced in the negative mode at m/z 567.1377, which was speculated to be C_25_H_28_O_15_; the parent ion lost C_3_H_6_O_3_,C_4_H_8_O_4_ and a glycosyl moiety, producing [M−H−C_5_H_9_O_4_−C_3_H_6_O_3_]^−^ at m/z 345, [M−H−C_5_H_9_O_4_−C_4_H_8_O_4_]^−^ at m/z 315 and [M−H−C_5_H_9_O_4_−C_6_H_11_O_5_]^−^ at m/z 272. This result was consistent with the results of a previous study [13]; it was thus identified as polygala xanthone III. The relevant MS/MS spectrum and fragmenting pathways are presented in Figure 6.

Component P10: a quasi-molecular ion [M+HCOOH−H]^−^ peak was produced in the negative mode at m/z 1123.5936, which was speculated to be C_53_H_90_O_22_; the parent ion lost a pentose moiety forming [M−H−Ara]^−^ at m/z 945, lost a glucosyl moiety forming [M−H−Glc]^−^ at m/z 915, lost a glycosyl moiety and a pentose moiety forming [M−H−Ara−Glc]^−^ at m/z 783, which further lost a glycosyl moiety forming [M−H−Ara−2Glc]^−^ at m/z 621, and which lost another glycosyl moiety forming [M−H−Ara−3Glc]^−^ at m/z 459. This result was consistent with those obtained in previous studies [14,15]; it was therefore identified as ginsenoside Rb_2_. The relevant MS/MS spectrum and fragmenting pathways are presented in Figure 7.

Component P20: a quasi-molecular ion [M−H]^−^ peak was produced in the negative mode at m/z 497.33, which was speculated to be C_31_H_46_O_5_; the parent ion formed [M−H−H_2_O]^−^ at m/z 479 and [M−H−CO_2_]^−^ at m/z 453 following dehydration and decarboxylation; the parent ion formed [M−H−C_2_H_5_COOH]^−^ at m/z 423 following a series of dehydration and decarboxylation reactions. This result was consistent those obtained in previous reports [20]; it was therefore identified as poricoic acid A. The relevant MS/MS spectrum and fragmenting pathways are presented in Figure 8.

### 2.6. Identification of Metabolites

Thirteen metabolites were identified in the serum of the normal group after the administration of KXS. The metabolic pathways included desugaring, hydroxylation, oxidation, hydration, deoxidation, demethylation, desaturation, glycine conjugation and their composite reactions. Nine metabolites were identified in the serum of the model group. The metabolic pathways included desugaring, hydration, oxidation, deoxidation, demethylation, desaturation, glycine conjugation, dehydration, acetylation and their composite reactions.

Component M1: a quasi-molecular ion [M−H]^−^ peak was produced in the negative mode at m/z 1143.6194, which was speculated to be C_54_H_92_O_23_; there were fragment ions at m/z 1107, m/z 945, m/z 783 and m/z 621 on the MS/MS spectrum, which were consistent with characteristic fragment ions of ginsenoside Rb_1_, and the mass difference was only 36 Da. Thus, component M1 was probably the metabolite of ginsenoside Rb_1_ after two hydration reactions. The metabolic pathway of ginsenoside Rb_1_ is presented in Figure 9.

Component M6: a quasi-molecular ion [M−H]^−^ peak was produced in the negative mode at m/z 501.3605, which was speculated to be C_31_H_50_O_5_; there were fragment ions at m/z 453 and m/z 439 on the MS/MS spectrum, which had a mass difference of 16 Da with fragment ions of tumulosic acid at m/z 437 and m/z 423. Thus, component M6 probably was the hydroxylated product of tumulosic acid. Similarly, component M4, M7 and M11 could be tentatively identified.

Component M14: a quasi-molecular ion [M−H]^−^ peak was produced in the negative mode at m/z 471.3515, which was speculated to be C_30_H_48_O_4_; there were fragment ions at m/z 423 and m/z 409 on the MS/MS spectrum, which had a mass difference of 14 Da with fragment ions of tumulosic acid at m/z 437 and m/z 423. Thus, component M14 probably was the demethylated product of tumulosic acid. Similarly, component M10 could be tentatively identified. According to m/z 389 of [M−H−C_6_H_10_]^−^, it was suggested that there be no demethylation in the parent nucleus; it was therefore presumed that its side chain lost a molecule of the methyl moiety. The metabolic pathways of tumulosic acid are presented in Figure 10.

Component M19: a quasi-molecular ion [M−H]^−^ peak was produced in the negative mode at m/z 453.3351, which was speculated to be C_30_H_46_O_3_; fragment ions at m/z 409 and m/z 391 on the MS/MS spectrum had a mass difference of 32 Da with fragment ions of poricoic acid G at m/z 441 and m/z 423, which probably consisted of two oxygen atoms; thus, component M19 probably was the deoxidized metabolite of poricoic acid G.

## 3. Discussion

AD is a chronic degenerative neurological disease, the clinical symptoms of which are memory loss, concentration loss, and dementia; the pathological manifestations are neuron loss, β-Amyloid protein precipitation, and entanglement of the nerve fibers caused by hyperphosphorylation of the tau protein [21]. The current well-accepted pathogenesis is β-Amyloid protein precipitation. A study found out that the AD rat model induced by the injection of Aβ_25–35_ into the hippocampal CA1 region on both sides was relatively stable within 8 weeks, and the procedure was repeatable and simple [22]. Therefore, the AD model in this work was induced by injecting D-galactose into the abdominal cavity for 4 weeks and subsequently injecting Aβ_25–35_ into the hippocampus on both sides.

To detect more transitional components in vivo, the present study carefully examined the timing of blood collection, which were 0.5, 1, 1.5, 2, 3, and 4 h after the administration in the preliminary experiment. The findings indicated that the TICs of 0.5 and 1 h after administration had more peaks, suggesting more components were detected. For the processing method of serum, methanol precipitation, acetonitrile precipitation and methanol-acetonitrile (1:1) precipitation were investigated. Results indicated that the TIC of serum processed by methanol-acetonitrile (1:1) precipitation had a better peak shape with relatively fewer interferences and a higher response. Thus, the serum obtained at 0.5 and 1 h after administration and methanol-acetonitrile (1:1) precipitation were finally chosen.

The transitional KXS components in vivo detected in the present study were mainly ginsenosides, triterpenoid acids of poria, polygala saponins, polygala xanthones and polygala ester, all of which were reported to be neuroprotective [23,24]. The metabolic pathways included phase I metabolism (hydroxylation, methylation, desaturation, oxidation, desugaring, dehydration, deoxidation and hydration), phase II metabolism (glycine conjugation, acetylation conjugation) and their composite reactions (Table 2, Figure 8 and Figure 9). The major transitional components of KXS in vivo were from ginseng radix et rhizome and polygalae radix [7]; in the current study, it was found out that the major transitional components of KXS in vivo were from poria, and their metabolic pathways were more diverse. This difference may be a result of different extracting methods or metabolic differences in rats. The relatively fewer metabolites of ginsenosides and polygala saponins were probably a hallmark of metabolism of saponins, namely, the ginsenosides and polygala saponins were readily hydrolyzed and desugared by relevant intestinal flora and enzymes, which then were absorbed into the blood in the form of aglycones. We also discovered that the contents of several aglycones were low in KXS; however, it was detectable in serum samples. Ginsenoside F_2_ was a desugaring metabolite of ginsenoside Rb_2_ [19]; given that the abundance of ginsenoside Rb_2_ in serum was relatively strong, it was reasonable to arrive at a preliminary conclusion that ginsenoside F_2_ was a metabolite of ginsenoside Rb_2_.

In the current study, nine common prototype components were detected in the rat serum of the normal and model groups after the intragastric administration of the KXS extract, including ginsenoside Rb_1_, Rc, Ra_1_, Rb_2_ and Rd, desacyl senega saponin B, poricoic acid A, poricoic acid B and tumulosic acid. The above nine components all have quite strong anti-AD activity as the major anti-AD components in KXS [15,25]. Furthermore, it was found out that the transitional components in the serum of normal rats were more abundant than those in the serum of model rats (See Table 2: 37 components for normal rats and 20 components for model rats). This finding is probably because the body has a weaker ability to absorb drugs and has decreased metabolic capacity under the pathological condition, leading to fewer components absorbed into the blood and a slower metabolic rate, so that some components are not detected in the serum samples. The absorption of drug components was significantly lower in model rats than in normal rats [26,27]. Zhou found out that the activity and genetic expression of human hepatic cytochrome P450 involved in Phase I metabolism and the activity of enzymes involved in Phase II metabolism underwent changes under the pathological conditions, leading to slowing down or speeding up of the drug elimination [26]. The processes of drug absorption, metabolism, distribution and excretion, namely ADME, are different between the normal and pathological states, which results in varying TCM components detected in different biological samples. Phase I and II metabolic reactions of drugs in the liver and other organs might be associated with chemical signals sent by endogenous components in different bodies, which further affects the metabolic transformation of drugs. Moreover, effective materials are used for the treatment of a particular disease; thus, a drug may have different effective materials for different diseases. This explains why studies using normal animals may fail to evaluate the efficacy of a drug comprehensively and precisely. Comparing the drug metabolism under the normal state and the AD pathological state, the current study was able to evaluate the effective material basis of KXS more appropriately.

Saponins are important active components of KXS; they tend to undergo phase I metabolism such as desugaring in the gastrointestinal tract, which is mostly associated with intestinal flora [28,29]. For instance, ginsenosides are rarely metabolized in the liver, but rather degrade by intestinal microbiota, producing more powerful metabolites [30]. Wang’s team [4] speculated that the components in polygala exert their activities through the metabolic transformation of the body. Guo [27] discovered that ginsenoside F1, ginsenoside Rh2, ginsenoside compound K, protopanaxatriol and other saponin metabolites can be detected in the serum of normal rats, but not in the serum of pseudo-sterilized rats. After modeling, the intestinal microecology of AD rats may have undergone some changes; for example, the composition and abundance of intestinal flora have changed, which further affects the degradation and absorption of some saponins, resulting in low levels and fewer kinds of these saponins. This led to the decrease in the levels of the component in the blood, which is consistent with our results, namely, higher levels of saponins were detected in the serum of normal rats than in the serum of model rats. Cao et al. [31] discussed effective material basis of KXS against depression based on the HPA axis (hypothalamic–pituitary–adrenal), suggesting that KXS may exert anti-AD action through the gut–brain axis. The conclusions of the above-mentioned studies [27,28,29,30,31] have verified our speculation that the gut–brain axis plays an important role in KXS working against AD, and our preliminary studies have also shown positive findings.

## 4. Materials and Methods

### 4.1. Materials

The following instruments were used in this study: Ultimate 3000 UHPLC (Thermo Fish, Waltham, MA, USA); LTQ-Orbitrap MS (including Xcalibur 2.1 workstation, Thermo Fish, Waltham, MA, USA); AE-240 electrical balance (Sartorius, Gottingen, German); SJIA5FE freezer dryer (Ningbo Shuangjia Instrument Co., Ltd., Ningbo, China); 118B high-speed grinder (Yongli Pharmaceutical Machinery Co., Ltd., Lishui, Zhejiang, China); supersonic cleaner (Kunshan Ultrasonic Instrument Co., Ltd., Kunshan, China); RE-52A rotary evaporator (Yarong Biochemical Instrument Co., Ltd., Shanghai, China); TGL-16.5M high-speed centrifuge (Shanghai Lu Xiangyi Centrifuge Instrument Co., Ltd., Shanghai, China); MTN-2800D nitrogen sample concentrator (Tianjin Automatic Science Instrument Co., Ltd., Tianjin, China); XH-B vortex mixer (Jiangsu Kangjian Medical Apparatus Co., Ltd., Taizhou, Jiangsu, China); automatic brain stereotaxic apparatus (RWD Life Science Co., Ltd., Shenzhen, China); TJ-1A micro-injection pump (Baoding Longer Precision Pump Co., Ltd, Buckinghan, Britain); miniature hand-held cranial drill (RWD Life Science Co., Ltd., Shenzhen, China); and Morris water maze (Noldus Information Technology Co., Ltd. Gelderland, The Netherlands).

Ginseng samples were purchased in Jian County, Jilin Province, which were authenticated as the dry root of *Panax ginseng* C.A. Mey. by Professor Zhi Liu from Jilin Agricultural University. Polygalae radix samples were purchased in Longhua Town, Yicheng County, Linfen, Shanxi Province. Poria samples were purchased in Gantang Town, Jingzhou County, Huanghua, Hunan Province. Acori tatarinowii rhizome samples were purchased from Shiji Pharmacy in Nanchang, Jiangxi Province. The above samples were authenticated as dry root of *Polygala tenuifolia Willd*, dry sclerotium of *Poria cocos (Schw.) Wolf* and dry rhizome of *Acorus tatarinowii Schott* by associate Professor Du Xiaolang from Jiangxi University of Chinese Medicine. All the samples were in accordance with pharmacopoeia standards, and voucher samples were deposited at the Key Lab of Modern Preparations of Traditional Chinese Medicine, Jiangxi University of Chinese Medicine, Nanchang, China.

Methanol, acetonitrile and formic acid were of chromatographic grade; all the other reagents were of analytical grade. Purified water was purchased from Wahaha Drinking water Co., Ltd. (Hangzhou, China). 

SPF-grade male SD rats (200 ± 20 g) with license No. SCXK (Xiang) 2019–0004 were purchased from Hunan STA Laboratory Animal Co., Ltd (Changsha, China). The rats were kept at room temperature of 24–24–26 °C, with relative humidity of 45–60% with 12 h of night-day alteration (7:00–19:00) and free access to food and water. The rats were allowed to acclimate to our animal facility for 7 d before the animal experiments were performed, which were approved by the Ethics Committee of Jiangxi University of Chinese Medicine (Approval number JZSYDWLL-20201115).

### 4.2. Method

#### 4.2.1. Preparation of 70% Ethanol Extract of KXS

A total of 120 g of ginseng sample, 120 g of poria, 80 g of polygalae radix and 80 g of acori tatarinowii rhizome samples were weighted; ginseng, poria and polygalae radix samples were passed through a No. 5 sieve (defined per the Chinese pharmacopoeia) and the acori tatarinowii rhizome sample was passed through a No. 2 sieve (defined per the Chinese pharmacopoeia). All sample powders were mixed well, then added to 8-fold 70% ethanol and soaked overnight. The sample solution was extracted under reflux for 3 h, followed by filtration while hot. The residue was extracted with 6-fold 70% ethanol under reflux for 1 h, followed by filtration while hot. The above filtrates were combined and concentrated at 65 °C until there was no ethanol dripping from the condenser [32]. The ethanol extract was freeze-dried, and 89.6 g of dry powder was obtained with a yield of 22.4%. This powder was mixed well, then packed and kept in a desiccator at room temperature for later use.

#### 4.2.2. Sample Solution

In total, 0.5 g of KXS powder was dissolved in 1 mL of 70% methanol. It was then vortexed for 15 min, followed by 30 min of ultrasonication. At last, the solution was centrifuged at 12,000 rpm for 10 min. The supernatant was acquired for MS analysis.

#### 4.2.3. AD Rat Model

The AD rat model was established according to the literature [22]: SD rats were divided into blank and model groups in random. The model rats were intraperitoneally injected with 50 mg·kg^−1^ of D-galactose, and blank rats were injected with equivalent normal saline for 4 consecutive weeks. Intracranial stereotaxic injections were performed on the 5th week: dry powder of Aβ_25–35_ was dissolved in normal saline and incubated in an incubator at 37 °C for 72 h for fibrosis. Totally, 1 mg·L^−1^ of oligomeric Aβ_25–35_ was obtained. Blank and model rats were anesthetized via intraperitoneal injection of 40 mg·kg^−1^ of 2% sodium pentobarbital solution; then, the rats were fixed on the stereotaxic apparatus with their heads steady after the hair in the brain region was removed using depilatory cream. Following sterilization of the scalp surface, it was cut open using a surgical knife along the cranial median line, separating the skin layers and dura maters until the skull was exposed. The drilling point was located at 3.0 mm behind the bregma and 2.0 mm beside the center line. A total of 4 μL of Aβ_25–35_ was vertically injected into 2.6 mm underneath the dura mater in both the hippocampal regions via a microsyringe. Blank rats were injected with equivalent normal saline. The injection rate was 0.8 μL·min^−1^. The needles stayed put for 5 min after the injection. Both holes were sealed with dental cement and then stitched. A total of 180,000 units of penicillin per rat was injected intramuscularly for 3 consecutive days.

#### 4.2.4. Indicator Detection

##### Observation of General Behavior

The changes in appearance, diet, and mental state of the rats were observed and evaluated during the experiments.

##### Morris Water Maze Test

The Morris water maze test was performed 1 week after the surgery [22]. The first stage was the orienting-navigation test. On day 1, the platform was fixed 1.5 cm under water in quadrant II. The rat was placed slowly into the water facing the pool wall, and allowed to find the platform freely within 60 s (time recorded); if the rat failed the test within 60 s, it was guided to the platform and allowed to stay for 20 s. The water maze test was done within 1 d, and the same test was repeated for 5 d. The second stage was the space-exploring test. The platform was removed on day 6, and the time that the rat spent in the target quadrant was within 60 s; the number of platform crossings and the swimming speed were recorded following the above operation.

##### Observation of Pathological Changes in the Hippocampal CA1 Region with Nissl Staining

Myocardial perfusion was performed with normal saline after the rat was sacrificed using anesthesia. Brain tissue was then collected and fixated in 4% polyformaldehyde solution, and the paraffin sections were deparaffinized and rehydrated. The sections were stained in toluidine blue staining solution for 1–2 min and washed with little water. The background was separated using 0.5% glacial acetic acid in a continuously separating and washing manner. The separating degree was controlled under the microscope. The sections were dried at 65 °C in an oven for over 4 h. They were put in clean xylene I for about 2 min, then in xylene II for 15 min. At last, the sections were mounted using neutral balsam and observed under an optical microscope at 400× magnification for pathological changes in the hippocampus.

#### 4.2.5. Drug Administration and Sample Collection

The model and blank rats were fasted but were allowed access to water 12 h before the experiments. Their own blank serum was collected as the control before drug administration. KXS freeze-dried powder was dissolved in 0.5% sodium carboxymethyl cellulose solution. Rats were intragastrically administered the drug at a dose of 10 g·kg^−1^ once a day for 7 consecutive days. At 0.5 h and 1 h after administration on the last day, orbital sinus blood was collected, which was then combined and centrifuged at 4000 rpm at 4 °C for 10 min. Following that, serum was collected and restored in the −80 °C freezer for later use.

#### 4.2.6. Handling of Serum Samples

A total of 300 μL of serum was added to 1.5 mL of a mixed solution of methanol-acetonitrile (1:1), which was then mixed using a vortex mixer at 2500 rpm for 15 min. The solution was sonicated for 10 min, and centrifuged at 10,000 rpm at 4 °C for 10 min. The supernatant was collected and blow-dried using nitrogen gas. The residue was re-dissolved in 100 μL of methanol, and then it was mixed with a vortex mixer at 2500 rpm for 2 min, followed by centrifugation at 12,000 for 10 min. The supernatant was collected, 2 μL of which was used for UPLC-MS analysis.

#### 4.2.7. Chromatographic Conditions

The separation was achieved on a ZORBAX RRHD Eclipse Plus C_18_ column (2.1 × 100 mm, 1.8 μm, Agilent Technologies, Inc., Santa Clara, CA, USA) with 0.1% formic acid in water (A) and acetonitrile (B) as the mobile phases. The gradient elution program was as follows: 0–14 min, 5–23% B; 14–30 min, 23–31% B; 30–50 min, 31–36% B; 50–56 min, 36–40% B; 56–62 min, 40–48% B; 62–76 min, 48–100% B; 76–77 min, 100% B; 77–77.1 min, 100% B -5% B. the flow rate was 0.3 mL·min^−1^, the injection volume was 2 μL and the column temperature was 40 °C.

#### 4.2.8. MS Conditions

The electrospray ion source was used as the ion source. Full mass/dd-MS^2^ scanning was performed in both the positive and negative modes within the range of m/z 100–2000. The other parameters were as follows: the resolution was 70,000 (FWHM), the capillary temperature was 300 °C, the sheath gas velocity was 35 L·h^−1^, the auxiliary gas flow rate was 10 L·h^−1^, the spray voltage was 3.6 V and the temperature of the ion transport tube was 320 °C. The fragmentation method used was collision-induced dissociation with a collision energy of 35 V.

#### 4.2.9. Data Handling and Analysis

The data of serum samples were pre-treated with the software SIEVE as follows: “Retention Time Start” was set as 0 min; “Retention Time Stop” was set as 80 min; M/Z range was set as 100–2000; “Frame time width” was set as 2.5 min; “M/Z Width” was set as 10 ppm; “Maximum Number of Frames” was set as 7000; and “Peak intensity Threshold” was set as 1000. The processed data were then imported into SIMCA 14.1 (MKS Umetrics, Umea, Sweden) to perform principal component analysis (PCA), in which the clustering of groups can be observed, and the differences between groups can be visualized. Subsequently, orthogonal partial least squares-discriminant analysis (OPLS-DA) and VLOOKUP function were employed to screen out the differential ions with VIP > 1.5, FC > 2 and *p* < 0.05 in the normal group and model group before and after dosing. Based on the previous recognition of chemical components in KXS (Appendix A) and the compound library of KXS (Appendix A), 34 metabolic types preset by “FISh Scoring and Background Removal” module in Compound Discoverer 3.0 were set to perform Phase I metabolism at the maximum of 3 stages and Phase II metabolism at the maximum of 1 stage in order to screen metabolites. Finally, these components were identified after integrating data from the previous KXS database and software Compound Discoverer.

The data of water maze were processed using SPSS 26.0 (SPSS Inc., Chicago, IL, USA); data of orienting-navigation were analyzed with two-factor repeated measures analysis of variances (ANOVA), and other data were analyzed using one-factor ANOVA. The results were represented as $\bar{\chi }$ ± s.

## 5. Conclusions

The current study successfully developed a stable and reliable rat AD model and an UPLC-LTQ-Orbitrap-MS method to analyze the prototype components and metabolites of KXS in rat serum. On this basis, the transitional components and metabolites of KXS in the serum of the normal and model rats were compared systematically. A total of 37 KXS-relevant components were identified in the serum of normal rats and 20 in the model rats, revealing the differences in the absorption/metabolism of rats with different pathological conditions. It was speculated that the intestinal microecological balance of the model rats was sabotaged, affecting the body’s absorption/metabolism of saponins, which further resulted in fewer transitional components in model rats than in normal rats. This study reflects the disposal of the drug by the body in a more objective manner, which contributes to the elucidation of material basis and mechanism of action of KXS against AD.

## Figures and Tables

**Figure 1 pharmaceuticals-16-00030-f001:**
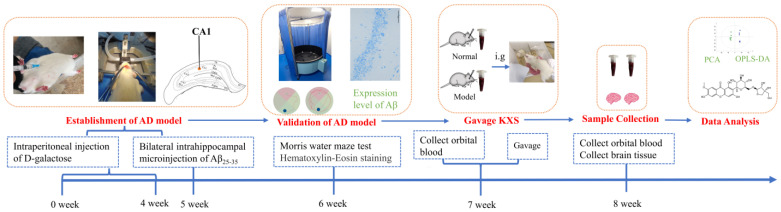
Schematic diagram of the study process.

**Figure 2 pharmaceuticals-16-00030-f002:**
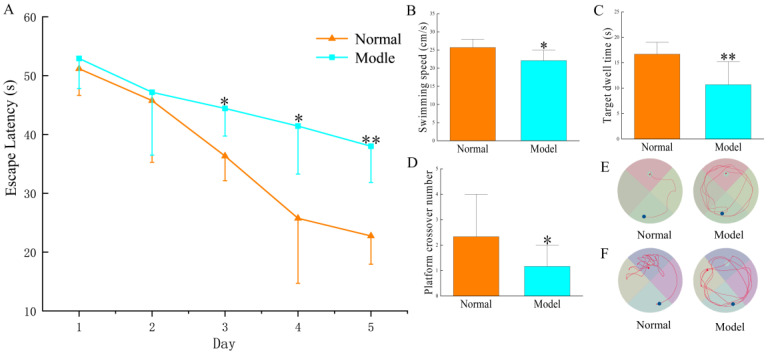
The Morris water maze test to evaluate spatial learning and memory ability. (**A**) escape latency in the water maze test; (**B**) swimming speed in the water maze within 60 s; (**C**) dwell time in quadrants of the platform within 60 s; (**D**) times of crossing the platform within 60 s; (**E**) swimming tracks on the 5th day of the navigation test; (**F**) swimming tracks in the space-exploring test. Note: compared with the normal group * *p* < 0.05, ** *p* < 0.01.

**Figure 3 pharmaceuticals-16-00030-f003:**
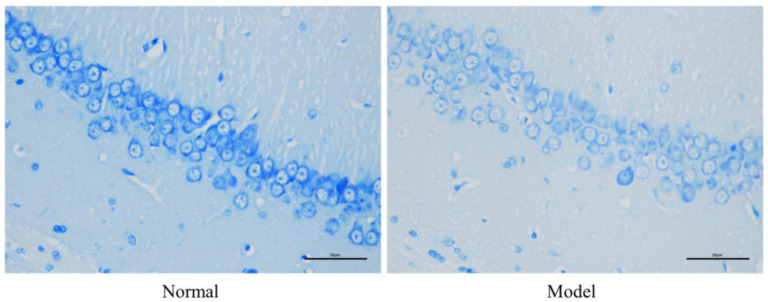
Neuronal structures in the hippocampal CA1 region of normal rats and model rats (Nissl staining ×400).

**Figure 4 pharmaceuticals-16-00030-f004:**
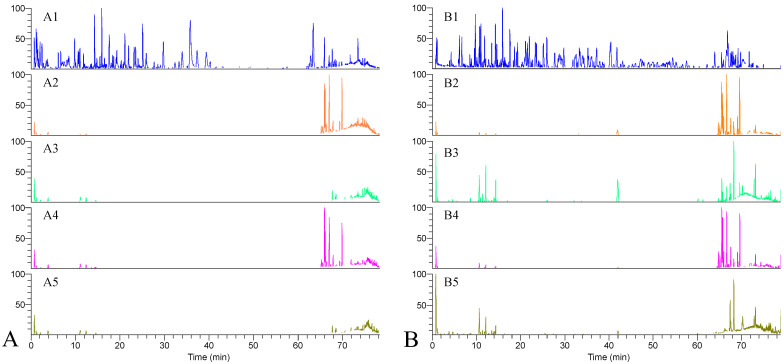
TIC of typical samples in positive (**A**) and negative (**B**) ion modes. 1. KXS extract; 2. blank normal group; 3. KXS-treated normal group; 4. blank model group; 5. KXS-treated model group.

**Figure 5 pharmaceuticals-16-00030-f005:**
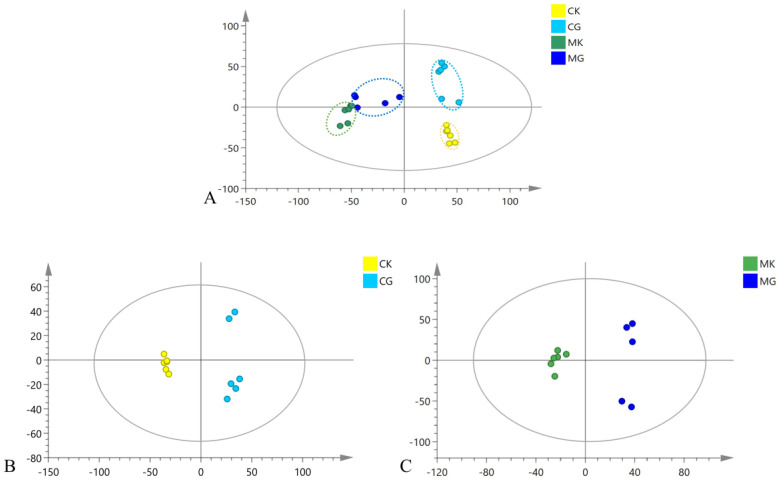
Multivariate statistical analysis of the serum metabolic profile of each group in the negative ion mode. (**A**) PCA plot of the CK, CG, MK and MG groups; (**B**) OPLS-DA plot of the CK and CG groups; (**C**) OPLS-DA plot of the MK and MG groups.

**Figure 6 pharmaceuticals-16-00030-f006:**
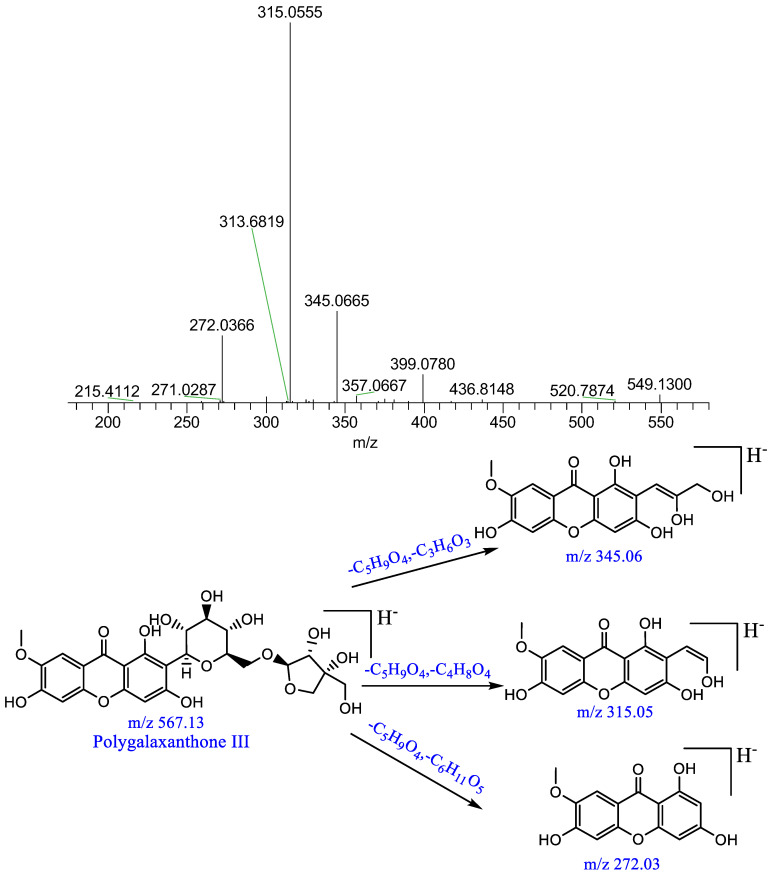
Mass spectrum of polygala xanthone III and possible fragmenting pathways.

**Figure 7 pharmaceuticals-16-00030-f007:**
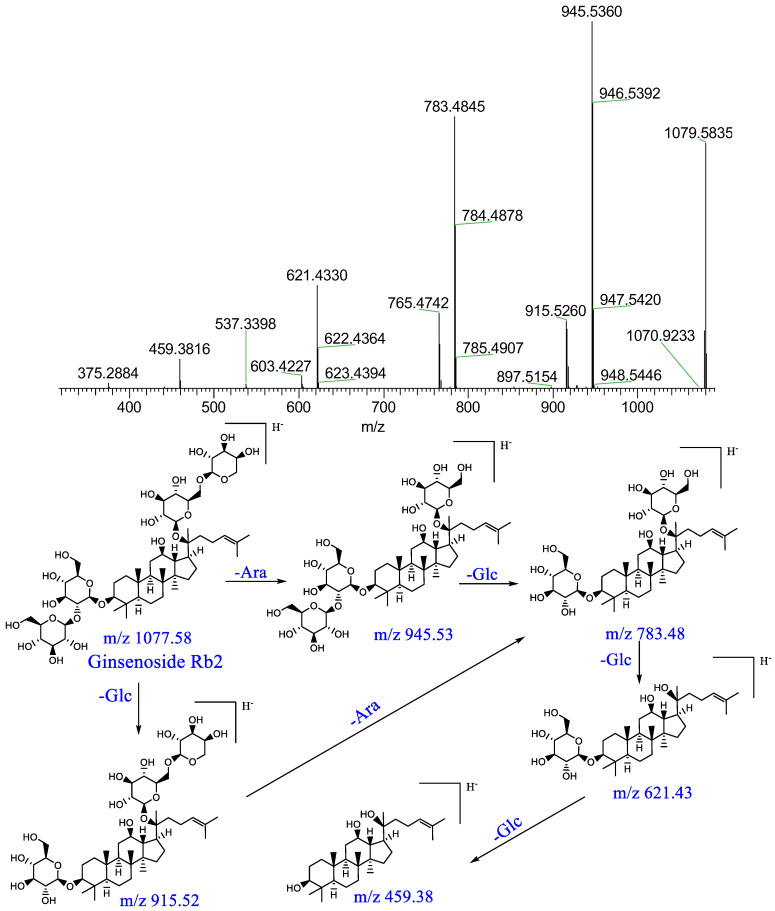
Mass spectrum of ginsenoside Rb_2_ and possible fragmenting pathways.

**Figure 8 pharmaceuticals-16-00030-f008:**
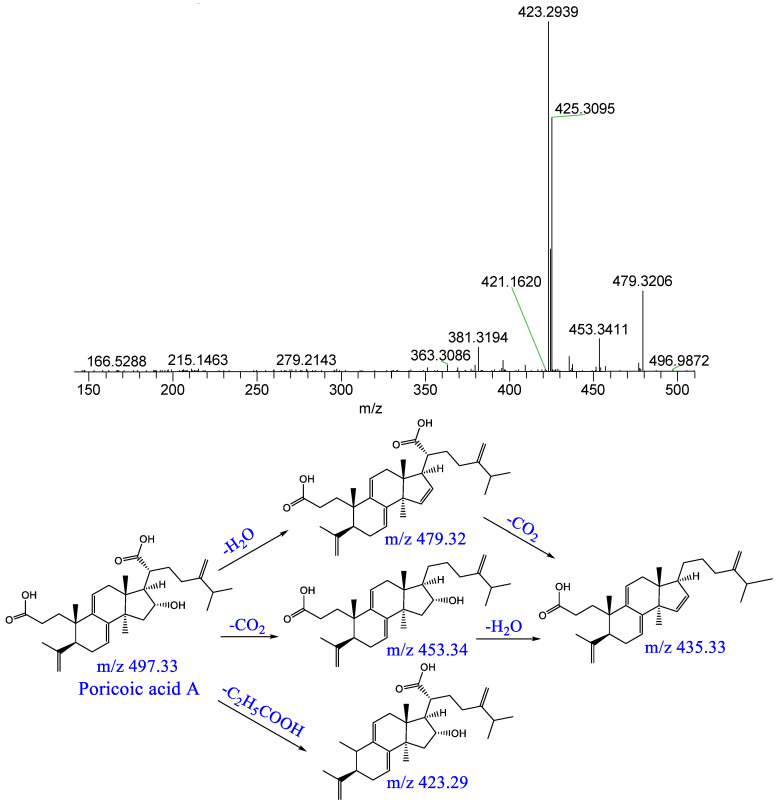
Mass spectrum of poricoic acid A and possible fragmenting pathways.

**Figure 9 pharmaceuticals-16-00030-f009:**
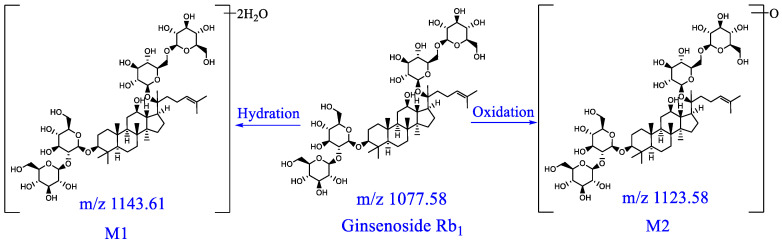
Possible metabolic pathways of ginsenoside Rb_1_ in rats in vivo.

**Figure 10 pharmaceuticals-16-00030-f010:**
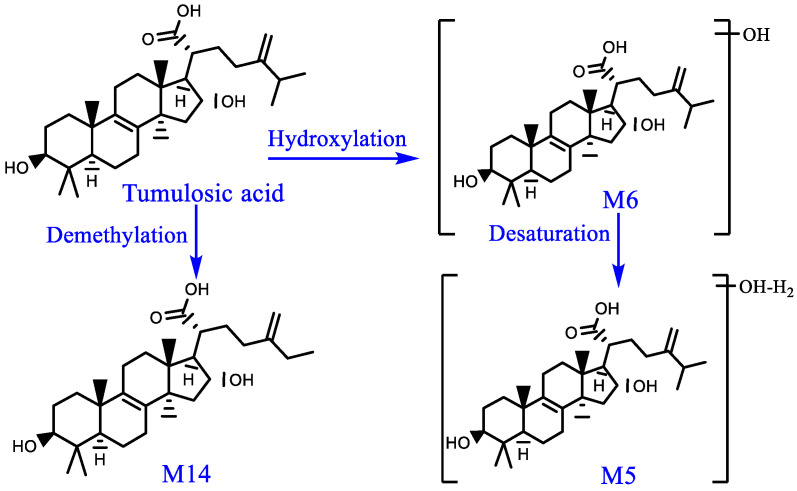
Possible metabolic pathways of tumulosic acid in rats in vivo.

**Table 1 pharmaceuticals-16-00030-t001:** Analysis and identification of prototype components in the rat serum of the KXS-treated normal group and model group.

No.	Compound	t_R_/min	Formula	Measured Mass (m/z)	Mass Error/ppm	Ion Addition	Fragment Ions (m/z)	* Origin	Group	Reference
P1	Polygalaxanthone VI	10.69	C_23_H_26_O_12_	493.1372	4.2	[M−H]^−^	317.0646, 302.0414, 175.0241	b	CG	[14]
P2	Polygalaxanthone III	10.88	C_25_H_28_O_15_	567.1408	9.3	[M−H]^−^	399.0780, 345.0665, 315.0555, 272.0366	b	CG	[6,15]
P3	Ginsenoside Rg_1_	17.42	C_42_H_72_O_14_	845.4913	2.3	[M−H+HCOO]^−^	799.4779, 637.4269, 619.4165, 475.3755	a	CG	[12,13]
P4	Ginsenoside Re	17.45	C_48_H_82_O_18_	991.5531	5.9	[M−H+HCOO]^−^	945.5341	a	CG	[12,13]
P5	Polygala saponin XXIX	20.34	C_64_H_102_O_33_	1397.616	−5.0	[M−H]^−^	1367.5997, 1173.5590, 1143.5489, 717.2394, 455.3132, 425.3030	b	CG	[14]
P6	Desacyl senega saponin B	21.04	C_59_H_94_O_29_	1265.5863	4.4	[M−H]^−^	1235.5588, 907.4617, 455.3132, 425.3029	b	CG/MG	[16]
P7	Tenuifoliose O	21.93	C_61_H_76_O_35_	1367.4114	1.4	[M−H]^−^	1337.5914, 1187.5401, 1143.5506, 1113.5404, 1011.5094, 455.3137, 425.3033	b	CG	[15]
P8	Polygala saponin XXII	22.23	C_58_H_92_O_28_	1235.5753	4.1	[M−H]^−^	1205.5500, 1011.5091, 555.1890, 469.1530, 455.3136, 425.3033	b	CG	[16]
P9	Polygala saponin XXVIII	22.95	C_53_H_84_O_24_	1103.5313	3.0	[M−H]^−^	1073.5091, 455.3137, 425.3034	b	CG	[15]
P10	Ginsenoside Rb_1_	31.77	C_54_H_92_O_23_	1153.6052	4.5	[M−H+HCOO]^−^	945.5345, 783.4833, 621.4320, 459.3811	a	CG/MG	[12,13]
P11	Ginsenoside Ro	32.6	C_48_H_76_O_19_	955.4943	3.6	[M−H]^−^	835.4423, 793.4320, 731.4323, 613.3702, 569.3808, 523.3756	a	CG	[12,13]
P12	Ginsenoside Rc	33.23	C_53_H_90_O_22_	1123.5935	3.6	[M−H+HCOO]^−^	945.5341, 783.4829, 765.4727, 621.4317, 459.3808	a	CG/MG	[12,13]
P13	Ginsenoside Ra_1_	33.51	C_58_H_98_O_26_	1255.6407	7.1	[M−H+HCOO]^−^	1077.5756, 945.5345, 915.5244, 783.4832, 621.4321	a	CG/MG	[6,16]
P14	Ginsenoside Rb_2_	34.93	C_53_H_90_O_22_	1123.5936	3.7	[M−H+HCOO]^−^	1079.5835, 945.5360, 915.5260, 783.4845, 765.4742, 621.4330, 459.3816	a	CG/MG	[12,13]
P15	Ginsenoside Rd	39.24	C_48_H_82_O_18_	991.5518	4.6	[M−H+HCOO]^−^	945.5352	a	CG/MG	[12,13]
P16	16α-hydroxytrametenolic acid	66.39	C_30_H_48_O_4_	471.3501	4.6	[M−H]^−^	450.9847, 425.3457, 409.3143, 407.3351, 339.2720, 337.2563	c	CG	[5,17]
P17	Poricoic acid B	66.81	C_30_H_44_O_5_	483.3135	4.1	[M−H]^−^	465.3050, 411.2939, 409.2783, 367.3037, 255.2351	c	CG/MG	[5,17]
P18	Dehydrotumulosic acid	67.49	C_31_H_48_O_4_	483.3503	4.9	[M−H]^−^	465.3412, 437.3461, 421.2759, 405.3196, 337.2564, 255.2350	c	CG	[5,17]
P19	Tumulosic acid	67.95	C_31_H_50_O_4_	485.3642	1.2	[M−H]^−^	437.3463, 423.3305, 389.2730, 337.2565, 275.204	c	CG/MG	[5,17]
P20	Poricoic acid A	68.13	C_31_H_46_O_5_	497.33	5.6	[M−H]^−^	479.3206, 453.3411, 435.3304, 425.3094, 423.2938, 409.2780, 381.3194	c	CG/MG	[5,17]
P21	Polyporenic acid C	69.33	C_31_H_46_O_4_	481.3296	−5.6	[M−H]^−^	481.3286, 463.3392, 437.2925, 419.2925, 403.2977	c	CG	[5,17]
P22	Poricoic acid Bisomer	69.4	C_30_H_44_O_5_	483.3151	7.4	[M−H]^−^	465.3056, 439.3259, 421.3151, 381.2835, 353.2518, 255.2351	c	MG	[5,17]
P23	3-epidehydrotumulosic acid	69.7	C_31_H_48_O_4_	483.3501	4.5	[M−H]^−^	421.3147, 391.2287, 255.2350	c	CG	[5,17]
P24	3β-hydroxylanosta-8,24-dien-21-oic acid	70.04	C_30_H_48_O_3_	455.3549	4.1	[M−H]^−^	455.17578, 434.9843, 372.2604, 338.2551, 297.2419, 279.2315	c	CG	[5,17]
P25	Poricoic acid A isomer	73.98	C_31_H_46_O_5_	497.3305	6.6	[M−H]^−^	455.3542, 437.3437, 401.2708	c	CG	[5,17]
P26	25-hydroxy-3-epitumulosic acid	74.02	C_31_H_50_O_4_	485.361	−5.3	[M−H]^−^	485.2794, 469.2485, 423.3243, 337.2518	c	MG	\

*: a, ginseng; b, *polygalae radix*; c, poria.

**Table 2 pharmaceuticals-16-00030-t002:** Analysis and identification of metabolites in the rat serum of the KXS-treated normal group and model group.

No.	Compound	t_R_/min	Formula	Measured Mass (m/z)	Mass Error/ppm	Ion Addition	Fragment Ions (m/z)	* Origin	Group	Reference
M1	Hydrated ginsenoside Rb_1_ (+2H_2_O)	31.73	C_54_H_96_O_25_	1143.6194	2.3	[M−H]^−^	1107.5870, 945.5357, 783.4841, 621.4331	a	CG	[7]
M2	Oxidated ginsenoside Rb_1_	32.57	C_54_H_92_O_24_	1123.5849	−4.9	[M−H]^−^	1098.5273, 1075.5377, 648.0545, 478.92459	a	CG	[7]
M3	Ginsenoside F_2_	55.98	C_42_H_72_O_13_	829.4915	−4.1	[M−H+HCOO]^−^	783.4844, 621.4329, 459.3815	a	CG/MG	[18]
M4	Dehydrotumulosic acid hydroxylation	59.54	C_31_H_48_O_5_	499.3472	8.8	[M−H]^−^	481.3288, 455.3499, 437.3394, 421.3083, 371.2567	c	CG	[19]
M5	Tumulosic acid hydroxylation + desaturation	59.87	C_31_H_48_O_5_	499.3395	−6.6	[M−H]^−^	481.3290, 453.3345, 437.3395, 421.3084, 371.2568, 313.2155	c	CG	\
M6	Tumulosic acid hydroxylation	61.08	C_31_H_50_O_5_	501.3605	3.9	[M−H]^−^	483.3526, 453.3416, 439.3624, 423.3309, 373.2783, 339.2361, 275.2043	c	CG	[19]
M7	Dehydrotumulosic acid hydroxylation	63.14	C_31_H_48_O_5_	499.3392	−7.2	[M−H]^−^	481.3292, 453.3343, 437.3032, 421.3085, 353.2465, 329.2104, 286.1924	c	CG	[19]
M8	Hydrated Pachymic acid	63.22	C_33_H_54_O_6_	527.3709	−6.0	[M−H-H_2_O]^−^	528.3739, 481.3656, 465.3344, 413.2672	c	MG	\
M9	Poricoic acid G dehydration + glycine conjugation	63.43	C_32_H_47_NO_5_	524.3392	2.0	[M−H]^−^	464.3157	c	MG	\
M10	Dehydrotumulosic acid demethylation	66.01	C_30_H_46_O_4_	469.3302	−4.4	[M−H]^−^	470.3327, 451.3188, 423.3241, 409.3085, 391.2983, 337.2517, 311.2000	c	CG	\
M11	16α-hydroxytrametenolic acid hydroxylation	66.06	C_30_H_48_O_5_	487.3406	−4.5	[M−H]^−^	469.3288, 439.3186, 425.3031, 397.3085, 355.2619, 287.2001, 207.1744	c	CG	[19]
M12	Dehydrotumulosic acid desaturation	66.65	C_31_H_46_O_4_	481.3383	1.2	[M−H]^−^	481.3455, 463.3396, 437.3393, 391.2593, 335.2360, 271.1690	c	CG	[19]
M13	Poricoic acid B Glycine conjugation	66.94	C_32_H_47_NO_6_	540.336	5.5	[M−H]^−^	480.3139, 409.2397	c	CG	[19]
M14	Tumulosic acid demethylation	67.29	C_30_H_48_O_4_	471.3515	7.6	[M−H]^−^	453.3416, 423.3309, 409.3151, 389.3576, 375.2575, 357.2467, 323.2410, 261.1883	c	CG/MG	\
M15	Poricoic acid G dehydration +deoxygenation + desaturation + glycine conjugation	73.06	C_32_H_49_NO_3_	494.3659	4.0	[M−H]^−^	433.3127, 295.3033, 196.0400	c	MG	\
M16	Poricoic acid G oxidation + glycine conjugation	73.66	C_32_H_49_NO_7_	540.328	−9.2	[M−H−H_2_O]^−^	481.2543, 255.2320	c	MG	\
M17	Oxidated pachymic acid	73.67	C_33_H_52_O_6_	525.3548	−7.0	[M−H−H_2_O]^−^	526.3585, 481.3656, 465.3344, 449.3034, 432.3006, 355.2257	c	MG	\
M18	Dehydrotumulosic acetylation	73.67	C_33_H_50_O_5_	525.3548	−7.0	[M−H]^−^	526.3585, 481.3656, 465.3344, 432.3006, 355.2257	c	MG	\
M19	Poricoic acid G deoxidation(−2O)	73.72	C_30_H_46_O_3_	453.3351	−5.0	[M−H]^−^	435.3237, 371.2565, 337.2515, 323.2360, 295.2261	c	CG/MG	\

*: a, ginseng; c, poria.

## Data Availability

Data are contained within the article and Appendix A.

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
