# Peer review of "A Comparative Study of Serum Pharmacochemistry of Kai-Xin-San in Normal and AD Rats Using UPLC-LTQ-Orbitrap-MS"

_pharmaceuticals, 2022, doi:10.3390/ph16010030_

Round 1
Reviewer 1 Report
Dear Author,
The work is beautifully organized and expressed in fluent language. However, it will be beneficial for readers and researchers to correct both the resolution and the text part of Figure 4 by checking it again. In addition, other figures in this text should be changed to high-resolution forms.
4.2.9. It is necessary to share the data of the data handling and analysis part in a separate supplementary file and to express this part in more detail.
I think that reviewing the conclusion part and organizing it in more detail will add efficiency to the study.
Regards
Author Response
- The work is beautifully organized and expressed in fluent language. However, it will be beneficial for readers and researchers to correct both the resolution and the text part of Figure 4 by checking it again. In addition, other figures in this text should be changed to high-resolution forms.
Response: Your comments and suggestions are appreciated. According to your requirements, Figure 4 has been re-plotted; the resolution of Figure 1, 2 and 5 has been increased from 300 dpi to 1000 dpi; the other Figures are imported original figures with satisfactory resolution, no changes can be made.
- 4.2.9. It is necessary to share the data of the data handling and analysis part in a separate supplementary file and to express this part in more detail.
Response: Supplementary material including Table S1 and Table S2 has been uploaded; and the corresponding part has been addressed in details under section 4.2.9 as follows:
Data of serum samples were processed by software SIEVE as follows: “Retention Time Start” was set as 0 min; “Retention Time Stop” was set as 80 min; M/Z range was set as 100-2000; “Frame time width” was set as 2.5 min; “M/Z Width” was set as 10 ppm; “Maximum Number of Frames” was set as 7000; “Peak intensity Threshold” was set as 1000. The processed data were imported into software SIMCA 14.1 to perform principal component analysis (PCA), in which the clustering of groups can be observed, and the differences between groups can be visualized. Subsequently, orthogonal partial least squares-discriminant analysis (OPLS-DA) and VLOOKUP function were employed to screen out the differential ions with VIP>1.5, FC>2 and P<0.05 in normal group and model group before and after dosing. Based on the previous recognition of chemical components in KXS (Supporting Information Table S1) and the chemical data base of KXS (Supporting Information Table S2), 34 metabolic types preset by “FISh Scoring and Background Removal” module in Compound Discoverer 3.0 were set to perform Phase I metabolism at the maximum of 3 stages and Phase II metabolism at the maximum of 1 stage in order to screen metabolites.
- I think that reviewing the conclusion part and organizing it in more detail will add efficiency to the study.
Response: Thanks. Changes have been made to the Conclusion section to achieve higher efficiency.
The current study successfully developed a stable and reliable rat AD model and an UPLC-LTQ-Orbitrap-MS method to analyze the prototype components and metabolites of KXS in rat serum. On this basis, the transitionalcomponents and metabolites of KXS in serum of normal and model rats were compared systematically. And a total of 37 KXS-relevant components in the serum of normal rats were identified, and 20 for model rats, revealing the differences in absorption\metabolismof rats in different pathological conditions. It was speculated that the intestinal microecological balance of model rats was sabotaged, affecting the body’s absorption\metabolism of saponins, which further resulted in fewer transitional components in model rats than those in normal rats. This study reflects the disposal of body to the drug in a more objective manner, which contributes to the elucidation of material basis and mechanism of action of KXS against AD.

Reviewer 2 Report
Point 1: In figure 4, the notations A, B, C in the title arenot correct.
Point 2: The author should explain more about the TIC results in section 2.2.
Point 3: In section 2.6, why are the metabolic pathways in the normal group and the model group different?
Author Response
Reviewer 2
- In figure 4, the notations A, B, C in the title arenot correct.
Response: Thank you for pointing that out. The notations has been re-arranged.
- The author should explain more about the TIC results in section 2.2.
Response: More about the TIC results has been added in section 2.2 as follows:
The ethanol extract of KXS and serum samples of blank normal group (CK), KXS-treated normal group (CG), blank model group (MK), KXS-treated model group (MG) were quantitatively analyzed under the conditions listed in Section 4.2.7 and 4.2.8. The total ion chromatograms (TIC) in positive and negative modes are shown in Figure 4. As it can be observed in Figure 4A1 and B1, KXS-relevant components achieved good separation within 80 min, and 160 principal components were identified based on our previous work [14,15]; relevant MS information was summarized in Table S1, which could be helpful for the identification of the components in serum. Comparing the TICs in both positive and negative modes (Figure 4A vs 4B), it could conclude that TIC in negative mode had better response, which was selected for the further MS analysis. Significant differences were observed comparing the intensity and number of peaks in serum chromatograms of normal group vs model group and those of before vs after dosing; however, components with low levels or response would readily be missed if their identification was based on the MS information of these differential peaks. In addition, the response of endogenous components was stronger than that of KXS-relevant components, interfering the identification of transitional components; hence, the in vivo prototype components and metabolites of KXS were recognized and identified with the assistance ofmultivariate statistical analysis (PCA and OPLS-DA).
Point 3: In section 2.6, why are the metabolic pathways in the normal group and the model group different?
Response: Thank you. Our findings showed that there are differences between the ADME in normal group and model group; however, these results could not be definitive proof for the conclusion that the metabolic pathways in the normal group and the model group are different. Thereby, the description that is relevant to “different metabolic pathways” has been removed from the revised manuscript. Referencing to some reports, the Discussing section has been revised as follows:
In the current study, 9 common prototype components were detected in rat serum of normal and model groups after intragastric administration of KXS extract, including ginsenoside Rb1,Rc,Ra1, Rb2 and Rd, desacylsenegasaponin B, poricoic acid A, poricoic acid B and tumulosic acid. The above 9 components all have quite strong anti-AD activity as the major anti-AD components in KXS [15,25]. Furthermore, it was found out that the transitional components in the serum of normal rats were more abundant than those in the serum of model rats (See Table 2: 37 components for normal rats and 20 components for model rats). This finding is probably because that under the pathological condition, the body has weaker ability to absorb drugs and decreased metabolic capacity, leading to fewer components absorbed into the blood and slower metabolic rate, so that some components are not detected in the serum samples. Some studies have shown that the absorption of drug components in model rats was significantly lower than that in normal rats [26,27]. Zhou found out that the activity and genetic expression of human hepatic cytochrome P450 involving in Phase I metabolism and the activity of enzyme involving in Phase II underwent changes under the pathological conditions, leading to slowing down or speeding up of the drug elimination [26]. The processes of drug absorption, metabolism, distribution and excretion, namely ADME, are different between normal state and pathological state, which results in varying TCM components detected in different biological samples. Phase I and II metabolic reactions of drugs in liver and other organs maybe associated with chemical signals sent by endogenous components in different bodies, which further affects the metabolic transformation of drugs. What’s more, effective materials are for the treatment of a particular disease, thus a drug may have different effective materials for different diseases. This explains why studies using normal animals may fail to evaluate the efficacy of a drug comprehensively and precisely. Comparing the drug metabolism under normal state and AD pathological state, the current study was able to evaluate the effective material basis of KXS more properly.
Saponins are important active components of KXS; they tend to undergo phase I metabolism such as desugaring in gastrointestinal tract, which mostly is associated with intestinal flora [28,29]. For instance, ginsenosides are rarely metabolized in the liver, but rather degrade by intestinal microbiota, producing more powerful metabolites [30]. Wang’s team [4] speculated that the components in polygala exert their activities through the metabolic transformation of body. Guo [27] discovered that ginsenosideF1, ginsenoside Rh2, ginsenoside compound K, protopanaxatriol and other saponin metabolites can be detected in the serum of normal rat, but not in the serum of pseudo-sterilized rats. After modeling, the intestinal microecology of AD rat may have undergone some changes, for example, the composition and abundance of intestinal flora have changed, which further affects the degradation and absorption of some saponins, resulting in low levels and fewer kinds of these saponins. This led to the decrease in the levels of component in the blood, which is consistent with our results, namely, the saponins detected in the serum of normal rats were more than those of model rats. Cao et al [31] discussed effective material basis of KXS against depression based on HPA axis (Hypothalamic-pituitary-adrenal), suggesting that KXS may exert anti-AD action through gut-brain axis. The conclusions of above studies [27-31] have verified our speculation that gut-brain axis plays an important role in KXS working against AD, and our preliminary studies have also achieved positive findings.

Round 2
Reviewer 2 Report
All the concerns are response.